# Modelling of amino acid turnover in the horse during training and racing: A basis for developing a novel supplementation strategy

R. Hugh Dunstan[1☯]*, Margaret M. Macdonald[1☯], Brittany Thorn[1‡], David Wood[2‡], Timothy K. Roberts[1‡]

**1** School of Environmental and Life Sciences, University of Newcastle, Callaghan, Australia, **2** Horsepower Pty Ltd, Windsor, NSW, Australia

☯ These authors contributed equally to this work.
‡ These authors also contributed equally to this work.
* hugh.dunstan@newcastle.edu.au

**Data Availability Statement:** All relevant data are within the paper and its Supporting Information files.

## Abstract

Horses in heavy training in preparation for racing and competition have increased metabolic demands to support the more intensive levels of exercise and recovery. However, little is known at the metabolic level about amino acid turnover and the specific alterations of demand caused by high intensity exercise. During exercise, certain amino acids are required in greater quantities due to disproportionate losses via excretory systems and usage in biosynthetic pathways. This investigation has built a theoretical computer model in an attempt to bring together the published rates of protein intake and utilisation to try to understand how some amino acids might be in higher demand than others. The model indicated that after evaluation of the daily amino acid turnover, glutamine/glutamic acid (Glx), serine and ornithine were in negative nitrogen balance which identified these amino acids as critical limiting factors for anabolism. Adjustment of the modelling conditions to cater for high intensity training indicated that an additional demand was placed on eight amino acids, including GLx, valine, lysine, histidine and phenylalanine which could thus become limiting under these conditions. The modelling results indicated that an amino acid supplement with the correct amino acids to match demand could theoretically be beneficial to a 500Kg horse in quantities of 20-80g/day. These results open new avenues of research for specifically tailoring amino acid supplementation to meet demands for sports horses in heavy training and improving general well-being, especially in hotter climates.

## Introduction

Assessment of the nitrogen balances of individual amino acids to establish whether they may be in deficit or in positive balance is critical to determining whether supplementation of specific amino acids may provide physiological benefits in horses. This approach has been successfully developed in the human context where it was found that histidine, serine, glycine and ornithine were used at disproportionately faster rates than the other amino acids during

**Funding:** The work was supported by the Gideon Lang Research Foundation. Dr David Wood works as a veterinarian consultant to Horsepower Pty Ltd to provide independent advice. Horsepower Pty Ltd provided support in the form of salary for author DW, but did not have any additional role in the study design, data collection and analysis, decision to publish, or preparation of the manuscript. The specific role of this author is articulated in the 'author contributions' section. Dr David Wood works as a veterinarian consultant to Horsepower to provide independent advice. Dr Wood assisted us in the final stages of preparation of the manuscript and provided valuable insight for checking the protein model and clinical interpretation of the data.474 The university authors have not been paid as consultants and have not received any direct grant funding from Horsepower Pty Ltd. This does not alter our adherence to PLOS ONE policies on sharing data and materials. The work was supported by the Gideon Lang Research Foundation. The funders had no role in study design, data collection and analysis, decision to publish, or preparation of the manuscript.

**Competing interests:** Horsepower Pty Ltd represent an Australian industry partner with an interest in developing effective supplements in the equine industry. This company had no role in study design, data collection and analysis, decision to publish, or preparation of the manuscript. Dr David Wood works as a veterinarian consultant to Horsepower to provide independent advice. Dr Wood assisted us in the final stages of preparation of the manuscript and provided valuable insight for checking the protein model and clinical interpretation of the data.474 The university authors have not been paid as consultants and have not received any direct grant funding from Horsepower Pty Ltd. This does not alter our adherence to PLOS ONE policies on sharing data and materials. Dr David Wood works as a veterinarian consultant to Horsepower to provide independent advice. Dr Wood assisted us in the final stages of preparation of the manuscript and provided valuable insight for checking the protein model and clinical interpretation of the data.474 The university authors have not been paid as consultants and have not received any direct grant funding from Horsepower Pty Ltd. This does not alter our adherence to PLOS ONE policies on sharing data and materials. The work was supported by the Gideon Lang Research Foundation. The funders had no role in study design, data collection and analysis, decision to publish, or preparation of the manuscript.

exercise and were thus potentially limiting under periods of high intensity training[1]. Horses excrete different proportions of amino acids in their sweat and urine compared with humans and thus require a specific approach to developing an equine model to evaluate nitrogen balance for amino acids. The study of nitrogen intake requirements has been researched extensively for horses and humans and rates of utilisation and excretion have been well defined [2–5]. However, assessment of the differential utilisation of ingested proteins appears to have only been undertaken in humans, where 15% of protein intake was oxidised for metabolism and 75% of protein intake was used for protein synthesis [2]. Furthermore, specific details on amino acid utilisation in horses are not readily available despite being required for more detailed evaluations of the nitrogen balance in terms of amino acids. Feeding rates have been established for various categories of light, medium, heavy and very heavy work and there is a vast array of high quality, commercially available feeds [6]. In the effort to find a competitive edge for maintaining top health condition and performance capacity, some work has been dedicated to investigating supplementation programs that would help maintain muscle mass and assist recovery rates from high intensity exercise in horses [7] and humans [3,8].

To set the context of daily supplementation regimes in regards to the daily protein requirements for the horse, it was determined that the recommended intake of protein for the horse undergoing a heavy exercise training program with racing was 860g crude protein/day for a 500Kg horse [6]. The corresponding rate of protein turnover in the muscle would generate in excess of 2,000g protein amino acids for synthesis of new protein and oxidation [9]. Horses provided with 35g/day of an amino acid mix comprising of 20g lysine and 15g threonine for 14 weeks under regular light exercise, responded by being able to better maintain muscle mass, had higher levels of creatinine and lower levels of the muscle damage marker 3-methylhistidine compared with the control group, irrespective of age [7]. It is difficult to understand why 20–40 g/day of an amino acid supplement, representing less than 5% of the daily available resources for oxidation and protein synthesis, could benefit the horse in training. A more detailed evaluation of nitrogen balance with a focus on amino acid turnover is thus required to understand the specific demands on these resources during high intensity exercise.

During the period of exercise, demands on nutrient resources increase to provide the necessary energy and metabolism required to support muscle exertion and recovery. The blood represents a reservoir of these resources which are distributed to muscles and interstitial fluids via an enhanced rate of circulation. Substantial quantities of amino acids can be lost through sweat which can represent up to 40–70% of the plasma reservoir quantity of amino acids during an exercise session [10], adding an additional demand on replenishment. During exercise, catabolism of muscle proteins occurs to release amino acids to restock the amino acids in circulation as well as feed the alanine-glucose cycle for gluconeogenesis. The rates of utilisation of amino acids for metabolism and their rates of loss in sweat are not all the same and some, such as serine, are lost at disproportionately faster rates than other amino acids [10]. Blood supply is reduced to the splanchnic region during exercise to provide further support to the muscles. Once high intensity exercise stops, the blood supply slowly returns to the splanchnic region but digestion is impaired for some hours following exertion [9,11–13]. During the initial recovery phase, the horse relies on the catabolism of muscle proteins to supply amino acid resources to support metabolism and repair processes [9,14–18].

Certain "non-essential" amino acids such as serine and glycine may be utilised at faster rates than they can be synthesised during exertion and recovery, thus becoming temporarily conditionally essential. In addition, some essential amino acids such as histidine may be oxidised or excreted at higher rates than others and must be replenished in the diet. Several amino acids have numerous roles in different metabolic pathways including energy metabolism and generating important complex metabolites, neurotransmitters and hormones [1,19].

The specific rates of utilisation of these amino acids are currently difficult, if not impossible, to accurately assess.

This investigation used published rates of protein intake, turnover and excretion and extrapolated these to individual appraisals of amino acids in sports horses [6,20]. A model was established to quantify the utilisation profile of amino acids by evaluating the average relative abundances of six dietary protein sources for nitrogen ingestion and then averaging five key proteins from the body to determine the estimated contributions of amino acids from protein turnover. The initial hypothesis was that by using extrapolated values for amino acid composition from the published rates for protein ingestion, turnover and excretion, a close to zero, positive balance should be achieved if it were assumed that amino acids were lost to metabolism and excretion pathways in the same composition profile as they occur in ingested and endogenous proteins. This would provide confirmation that the nitrogen turnover model has been correctly constructed. Subsequent testing of the model would then involve using published rates of amino acid losses in urine and sweat (including latherin composition) where certain amino acids are lost at disproportionately faster rates in horses than other amino acids. This would test the hypothesis that specific losses via excretion pathways could alter the nitrogen balances of certain amino acids, resulting in increased demands for their provision. In this way, it was possible to estimate potential gram quantities of deficits of key amino acids that might be critical to maintaining a positive nitrogen balance and therefore improving general health and performance for the horse.

## Materials and methods

Recommendations for protein intakes for horses are largely dependent on the magnitude of work they are undertaking. Four categories of work have been described which include light, moderate, heavy and very heavy [6,20]. This model has been developed in the context of a 500Kg sports horse undertaking a "heavy" workload for training and racing as defined by the NRC, 2007. The nitrogen balance was calculated on the basis that

Nitrogen balance gN/Kg BW/day = Nitrogen intake gN/Kg BW/day–Nitrogen losses gN/Kg BW/day [21]

This has been extensively investigated in humans and horses and well summarised in a number of publications [2,3,9,20,22,23]. These nitrogen flux assessments have been converted to grams of protein by applying a conversion factor of 6.25g protein per g of nitrogen [2]. The recommended protein intake rate was 1.72g/Kg BW/day for the horse undertaking a heavy work schedule [6] which was similar to the recommended rates for human athletes at 1.2–1.7 g protein/kg/day [3]. Grams of amino acids were extrapolated from grams of protein from a range of different plant sources (Table 1).

The horse has been shown to synthesise 3–5 times more protein than the amount ingested on a daily basis [9,23] and thus they rely heavily on the endogenous turnover of body proteins to meet demand [9,22]. The next challenge was to determine the amino acid profiles generated by the catabolism of a broad range of body proteins. It cannot be assumed that all proteins have similar amino acid compositions. Some key proteins have very high requirements of specific amino acids for biosynthesis, such as histidine, leucine and valine for haemoglobin, and glycine and proline for collagen. For the purposes of developing an initial simplistic model, these protein profiles of amino acids were averaged to generate a set of percentage abundances for amino acids generated from the catabolism of body proteins (Table 1). It was estimated that around 30% of the total protein mass in mammals is collagen [24] and muscle proteins comprise around 42–52% of the body protein in horses[25]. These two protein groups were chosen for our model not just because of their large body component, but because they are

**Table 1. The average percentage relative abundance compositions of selected amino acids in body protein composition and dietary sources[1].**

Average % contents of key amino acids in body proteins and dietary proteins

| Dietary | His | Ser | Gly | Lys | Asp | Glx | Leu | Ile | Val | Thr | Met | Tyr | Phe | Pro | Ala |
|---|---|---|---|---|---|---|---|---|---|---|---|---|---|---|---|
| Legume | 2.2% | 5.1% | 5.4% | 6.0% | 12.2% | 11.2% | 8.6% | 4.8% | 5.8% | 5.1% | 1.5% | 3.6% | 5.3% | 11.4% | 6.6% |
| Corn | 2.7% | 4.9% | 4.7% | 3.0% | 6.5% | 21.0% | 11.0% | 3.9% | 4.7% | 2.6% | 1.4% | 4.0% | 4.9% | 8.9% | 7.8% |
| Oats | 2.3% | 4.3% | 5.0% | 4.3% | 9.1% | 24.6% | 7.6% | 4.0% | 5.1% | 3.4% | 2.6% | 3.2% | 5.4% | 4.8% | 5.1% |
| Lucerne | 3.1% | 5.2% | 4.6% | 6.0% | 19.1% | 9.4% | 7.5% | 4.2% | 6.0% | 5.1% | 0.8% | 4.6% | 4.5% | 9.0% | 6.0% |
| Barley | | 6.4% | 6.5% | 2.6% | 6.0% | 23.0% | 7.8% | 3.7% | 6.6% | 4.3% | 1.4% | 2.1% | 4.8% | 15.9% | 6.4% |
| Wheat | 2.0% | 5.4% | 6.4% | 2.0% | 4.1% | 34.9% | | 4.3% | 5.2% | 3.0% | 1.7% | 1.9% | 4.7% | 15.4% | 4.6% |
| **Plant average** | **2.5%** | **5.2%** | **5.4%** | **4.0%** | **9.5%** | **20.7%** | **8.5%** | **4.2%** | **5.6%** | **3.9%** | **1.6%** | **3.2%** | **4.9%** | **10.9%** | **6.1%** |
| **Body** | **His** | **Ser** | **Gly** | **Lys** | **Asp** | **Glx** | **Leu** | **Ile** | **Val** | **Thr** | **Met** | **Tyr** | **Phe** | **Pro** | **Ala** |
| Albumin | 3.5% | 3.4% | 1.4% | 11.8% | 9.4% | 15.0% | 10.4% | 2.4% | 5.4% | 5.2% | 0.7% | 4.9% | 5.9% | 4.4% | 4.9% |
| Collagen | 0.6% | 3.9% | 35.0% | 3.5% | 4.2% | 7.1% | 2.2% | 1.3% | 1.4% | 1.3% | 0.8% | 0.3% | 0.8% | 22.9% | 9.6% |
| Haemoglobin | 8.3% | 4.1% | 4.3% | 9.3% | 9.6% | 6.6% | 13.9% | | 10.4% | 5.1% | 1.3% | 3.1% | 7.3% | 5.0% | 9.2% |
| Actin | 2.0% | 6.4% | 7.5% | 5.0% | 9.4% | 10.9% | 6.9% | 7.3% | 5.0% | 7.5% | 4.4% | 4.3% | 3.1% | 5.0% | 8.0% |
| Myosin | 1.8% | 4.5% | 4.6% | 10.4% | 10.4% | 18.7% | 10.1% | 3.6% | 4.2% | 4.7% | 2.4% | 1.8% | 3.1% | 3.2% | 9.1% |
| **Mean** | **3.2%** | **4.5%** | **10.6%** | **8.0%** | **8.6%** | **11.7%** | **8.7%** | **3.6%** | **5.3%** | **4.8%** | **1.9%** | **2.9%** | **4.0%** | **8.1%** | **8.2%** |
| Average of Hb and myosin | *5.1%* | *4.3%* | *4.5%* | *9.8%* | *10.0%* | *12.6%* | *12.0%* | *3.6%* | *7.3%* | *4.9%* | *1.8%* | *2.4%* | *5.2%* | *4.1%* | *9.1%* |
| The percentage abundances were adjusted to compensate for an elevated demand for Hb and myosin during training / racing *(profile used in Table 5, column G)* | | | | | | | | | | | | | | | |
| Adjusted amino acid demands to cater for Hb and myosin synthesis[2] | **4.2%** | 4.5% | **6.6%** | **9.0%** | **9.6%** | **12.7%** | **9.7%** | 3.6% | **6.3%** | 4.8% | 1.9% | 2.9% | **5.0%** | **5.1%** | 8.2% |

[1] Plant proteins: legume [32], corn [33], oats [34], lucerne [35], barley [36] and wheat [37]

Human/animal proteins: collagen [**38**], albumin [39], haemoglobin [40,41], actin [42,43] and myosin [43].

[2] It was argued that an increased demand for synthesis of these proteins would temporarily skew utilisation profiles because, combined, these proteins have higher requirements for histidine, lysine, aspartic acid, Glx, leucine, valine and phenylalanine and lower requirements for glycine and proline. The percentage relative abundances for these components were thus each adjusted upwards by 1% and glycine and proline adjusted downward by 4% and 3% respectively to compensate.

most likely to undergo an exacerbated turnover to facilitate growth, repair and development during the course of fitness training [3,26,27]. Similarly, there will be a requirement of enhanced erythropoiesis to support the demand for increased aerobic capacity [28–30]. Albumin synthesis rate was estimated based on human production rates of 0.2g/Kg BW to be around 5% of daily protein synthesis [31]. The precise demands on endogenous protein turnover to supply amino acids for metabolism and repair will likely vary depending on the stage of training. The model was thus conceived to ascertain an "average" composition of amino acids from these five key proteins to begin an understanding of what might influence amino acid balances during heavy training.

In the context of assessing the metabolism of amino acids, it cannot be assumed that amino acids will be metabolised or excreted in the same proportions that they are ingested. This is because certain amino acids such as serine, glycine and histidine are used in multiple biochemical pathways for metabolism and biosynthetic formation of key products (eg carnosine, phospholipids, purines and porphyrin rings). The non-essential amino acids can be made endogenously and would thus skew the percentage abundances of ingested amino acids. Some amino acids are lost at disproportionately faster rates than other amino acids in the sweat and urine. Since it was not feasible to assess the relative metabolic fluxes of the amino acids in this context, it was proposed that the best estimates would be obtained by setting the rates of utilisation in line with the average amino acid compositions determined from the five significant body proteins presented in Table 1. The parameters used to define the model have been converted to grams of protein or grams amino acids/Kg BW/day as appropriate and summarised in Table 2.

**Table 2. Summary of the parameters used in the modelling of amino acid fluxes based on protein intake, turnover, metabolism and excretion.**

| Parameter | | Rationale |
|---|---|---|
| Body weight of horse | 500Kg | Recommendation for "heavy" work[6] |
| Sources of protein for metabolism: | 1.72g/Kg BW/day | Calculated as 4g/Kg BW/day for horse[44] |
| Ingested protein | 4g/Kg/day | 15-20g per 100–110 g dietary intake |
| Endogenous protein turnover | 15% | 70-80g per 100-110g dietary intake |
| Utilisation of protein resources: | 75% | 80g per 300g proteins turned over |
| % of protein intake directly oxidised for metabolism | 27% | 206-211g per 300g proteins turned over [2] |
| % of protein intake used for protein synthesis | 67% | • 202 mg protein/Kg BW/day from urine, |
| % of protein turnover oxidised for metabolism | 372mg/Kg BW/day | • 41 mg protein/Kg BW/day from faeces |
| % of protein turnover used for protein synthesis | | • 129 mg protein/Kg BW/day from skin, sweat proteins and sweat fluid[10,20,44,45] |
| Excretion of protein resources: | | |
| Obligatory N excretion losses as protein equivalents: | | |
| To extrapolate protein flux to amino acids: | | Average plant composition (Table 1) |
| 1) Average amino acid compositions of protein sources were determined from: | % | Average body composition (Table 1) |
| • Ingested plant proteins averaged from 6 potential sources | % | Average plant composition (Table 1) |
| • Endogenous protein turnover averaged from 5 key body proteins | % | Average body composition (Table 1) |
| 2) The individual rates of utilisation of amino acids were then extrapolated from the protein values in terms of oxidation and protein synthesis | % | Sweat: Profiles integrated from inclusion of skin, fluid and protein contributions [10,44,46,47]B; 1.0% BW/day[20] |
| • Ingested proteins *(15% oxidised; 75% for protein synthesis)* | | Urine: Profiles from Table 4; 153mg N/Kg BW$^{0.75}$/day [9,10,20,44] |
| • Endogenous protein turnover *(27% oxidised; 67% for protein synthesis)* | | Faeces: The amino acid profile was assigned as that representing the average body composition of key proteins in Table 1; 40.9 mg protein/Kg BW/day [9] |
| 3) Excretion of amino acids: | | |
| • Urine and sweat profiles were based on measured average compositions | | |
| • It was assumed that the amino acids were excreted in faeces in the same general proportions that they appear in the average body protein composition profile (Table 1) | | |
| • Separate contributions to excretion were calculated from ingested proteins and protein turnover | | |

The model was developed to estimate the nitrogen balances of amino acids shown to be lost in sweat in much higher concentrations than found in plasma: serine, glutamic acid/glutamine (Glx), histidine, phenylalanine and aspartic acid. Other amino acids were also included in the assessments because they were present in high abundances within the sweat: glycine, alanine, leucine, lysine, valine, proline, tyrosine and ornithine (non-proteinogenic). Two amino acids that were shown to be present in lower concentrations in the sweat relative to the plasma, were also included: threonine and methionine. These calculations enabled estimates of nitrogen balance for specific amino acids.

A scenario was proposed and tested whereby horses undergoing training to increase fitness levels will have an increased demand to enhance muscle mass and improve levels of haemoglobin synthesis and erythropoiesis [30]. To accommodate this, the average amino acid compositions of haemoglobin and myosin were determined and have been presented in Table 1. The nitrogen balances for amino acids were then calculated to determine potential alterations that might be expected to coincide with a high demand for haemoglobin and erythrocyte production during fitness training.

The proportions of amino acids excreted in urine were required to accurately integrate specific losses of amino acids in the model of amino acid turnover. Resting urine samples were also collected from four Standardbred horses in the early morning prior to training [10]. The animals were trained to urinate on command for race testing and a large clean container with an extended handle was placed into the stream of urine. Collected urine was transferred to a sterile Monovette® tube for preservation and transport to the laboratory for analysis. The amino acid composition of samples was determined *via* EZ:Faast™ derivatisation (Phenomenex Inc.) followed by GC/FID analysis. The EZ:Faast™ procedure consists of a solid phase extraction step, followed by derivatisation and a liquid/liquid extraction as described previously [10]. Approval was received from the University of Newcastle Animal Care and Ethics Committee (approval numbers A-2012-257).

## Results and discussion

The nitrogen balance model developed for assessing the turnover of amino acids was run for a 500Kg horse undertaking a heavy training load for racing. The protein intake rate was set at 1.72g/Kg BW/day and the endogenous protein turnover rate was set at 4g/Kg BW/day. Under the conditions established in Table 2, it was possible to calculate the quantities of protein utilised for oxidation and protein synthesis, together with the losses from excretion as shown in Table 3. It was apparent from this output that the total daily protein synthesis of 1,978g was three times the protein synthesis generated from protein intake (645g) as suggested by Martin-Rosset [9,44].

The average amino acid composition of the urine has been summarized in Table 4, where the results have been sorted on the basis of the most abundant to the least abundant components together with the relative percentage compositions. These data were required to provide the most realistic excretion profile data from the horse for use in modelling the nitrogen balance. The urine profile of amino acid composition has been compared with the average urine composition measured for 153 human subjects comprising 101 males and 52 females from a previous study [48]. The purpose for this comparison was to highlight the differences between the human and equine urine compositions of amino acids. The humans had nearly double the total concentration of amino acids in first morning urine samples compared with the equivalent horse samples. The higher concentrations in the human urine profile were largely catered for by the very high levels of histidine and glycine which were present at levels >1,000μmoles/L. This would be predicted to have a significant impact on the protein turnover modelling, leading to a different profile of amino acids required for replenishment in horses compared with that proposed for humans [1]. Histidine is essential for both species, but humans can derive larger quantities of this component via the high proportions of carnosine (β-alanyl-L-histidine) present in meat [1,49,50]. As herbivores, the horses seem to have developed a more efficient system of preventing losses of key amino acids such as histidine and glycine via urine excretion pathways (Table 4).

**Table 3. The daily utilisation of protein resources for a horse in heavy exercise training[1].**

| 500Kg Horse | Protein contributions g | Oxidised g | Protein synthesis g | Losses from urine, faeces, sweat g |
|---|---|---|---|---|
| Protein intake | 860 | 129 | 645 | 56 |
| Protein turnover contribution | 2,000 | 540 | 1,333 | 130 |
| Totals | 2,860 | 669 | 1,978 | 186 |

[1]These were calculated by assuming a protein intake of 1.72g protein/kg BW/day and fixed rates of utilisation by oxidation, protein synthesis and excretory losses, where values have been rounded for simplicity

**Table 4. Pre-exercise, pre-feeding urinary amino acid concentrations (n = 4) with corresponding percentage relative abundances for Standardbred horses.** Amino acids have been listed in order from the highest to lowest urinary concentrations[1].

| Amino Acid | Horse Pre-exercise Urine Concentration (μmole L$^{-1}$) ± SE | Percentage abundance | Human* Fasted first morning urine (μmole L$^{-1}$) ± SE | Percentage abundance |
|---|---|---|---|---|
| β-Aminoisobutyric acid | 436 ± 70 | 21% | 172 ± 19.8 | 3.4% |
| Glutamic acid | 394 ± 194 | 19% | 20 ± 1.6 | 0.4% |
| Glutamine | 188 ± 28 | 9.1% | 520 ± 23 | 10% |
| Glycine | 182 ± 30 | 8.8% | 1,051 ± 56 | 21% |
| Serine | 165 ± 65 | 8.0% | 330 ± 19 | 6.5% |
| α-Aminoadipic acid | 152 ± 9 | 7.3% | 57 ± 4.6 | 1.1% |
| Lysine | 79 ± 20 | 3.8% | 253 ± 34 | 5.0% |
| Ornithine | 69 ± 23 | 3.4% | 49 ± 10 | 1.0% |
| Threonine | 49 ± 11 | 2.4% | 133 ± 13 | 2.6% |
| Cystine | 42 ± 14 | 2.0% | 63 ± 3.1 | 1.2% |
| Histidine | 39 ± 7.9 | 1.9% | 1,122 ± 86 | 24% |
| Alanine | 36 ± 10 | 1.76% | 260 ± 14 | 5.1% |
| Allo-Isoleucine | 34 ± 13 | 1.7% | 0.1 ± 0.1 | 0.0% |
| Proline-Hydroxyproline | 28 ± 4.6 | 1.4% | 207 ± 12 | 4.1% |
| Hydroxylysine | 27 ± 7.5 | 1.3% | 61 ± 3.8 | 1.2% |
| Asparagine | 24 ± 8.0 | 1.2% | 212 ± 14 | 4.2% |
| Glycine-Proline | 20 ± 6.7 | 0.97% | 70 ± 4.2 | 1.4% |
| Methionine | 16 ± 6.6 | 0.81% | 12 ± 0.9 | 0.2% |
| Valine | 15 ± 4.4 | 0.74% | 45 ± 1.9 | 0.9% |
| Cystathionine | 15 ± 5.4 | 0.73% | 36 ± 2.9 | 0.7% |
| Phenylalanine | 14 ± 4.1 | 0.70% | 49 ± 2.3 | 1.0% |
| Tyrosine | 14 ± 2.3 | 0.70% | 89 ± 5.4 | 1.7% |
| α-Aminopimelic acid | 8.8 ± 3.5 | 0.43% | 33 ±19 | 0.6% |
| Leucine | 6.5 ± 1.5 | 0.31% | 31 ± 1.6 | 0.6% |
| Tryptophan | 4.0 ± 2.1 | 0.19% | 65 ± 4.4 | 1.3% |
| α-Aminobutyric acid | 3.0 ± 3.0 | 0.15% | 8.4 ± 0.9 | 0.2% |
| Aspartic acid | 2.7 ± 2.7 | 0.13% | 17 ± 1.7 | 0.3% |
| Proline | 2.3 ± 2.3 | 0.11% | 8.6 ± 1.0 | 0.2% |
| Sarcosine | 0 ± 0 | 0.00% | 0.1 ± 0.1 | 0.0% |
| Isoleucine | 0 ± 0 | 0.00% | 10 ± 0.9 | 0.2% |
| Total | 2,068 ± 142 | | 5,106 ± 245 | |

[1]These values have been compared with human equivalent data collected from a previous study from 101 males and 52 females [48]. The human data were recalculated to present the average concentrations found in the combined group of males and females.

The modelling program was designed to assess the final daily balances of amino acids for the horse in heavy work training by assessing protein intake rates, utilisation rates and excretion rates from various studies. This category of work was selected to represent the horses in preparatory stages for various types of competition and the specific demands will obviously vary depending on the breed of horse and the nature of events. The model was initially run on the assumption that the amino acids would be lost via excretion pathways in the same proportions that they appear in food and body proteins. Assessments of the use of 2,860 g of amino acids made available from ingested food and turnover of endogenous proteins, resulted in positive balances for all of the amino acids, with a total balance of 25.7g (Table 5). This was

interpreted to support the model's capacity to assess the nitrogen balance in terms of amino acid turnover.

It is known that in reality, the amino acids are not metabolized or excreted in the same ratios as they appear in ingested or endogenous proteins, as seen for example, from the high concentrations of glutamine/glutamic acid (Glx), glycine, serine and lysine present in the equine urine (Table 4). The model was then run by using the proportions of amino acids measured in urine and sweat (see S1 Example calculation). Most of the amino acids returned a positive nitrogen balance (H), but Glx, ornithine and serine had prominent negative nitrogen balances of -5.7g/day, -3.4g/day and -3.0g/day, respectively. Interestingly, several other amino acids such as leucine, aspartic acid, proline and alanine showed relative increases in the nitrogen balance, with an overall net balance of 77.2g/day. In a parallel study modelling protein turnover in humans, it was found that Glx was not in negative balance, but serine, ornithine, histidine and glycine were in both males and females [1].

Glutamine and glutamic acid (Glx) are lost in high proportions in equine urine whereas serine is the most abundant amino acid in equine sweat [10] which would be consistent with their negative balances generated in this model. Serine is by far the most abundant free amino acid in horse sweat fluid where it has been measured in Standardbred horses at 791–893 μmol/L which is 4–5 times higher than corresponding levels measured in horse plasma [10]. The

**Table 5. Summary of the estimated daily levels (grams) of amino acid intake, turnover and excretion for a 500Kg horse undertaking a heavy workload.** The balance of amino acids was first calculated assuming that amino acids were excreted in the same proportions that they appear in food and body proteins (F). The balance was then calculated using actual proportions of amino acids measured in sweat and urine (H).

| Amino acid (AA) calculated on the average composition in food proteins or turnover of body proteins (Table 2) | (A) Amino acids from 860g protein intake /day[1] = % $AA_{Food}$ x 860 g/day | (B) Amino acids from 2,000g endogenous protein turnover /day[2] = %$AA_{body}$ x 2000 g/day | (C) Usage of amino acids for protein synthesis[3] = —(75%xA)— (67%xB) | (D) Usage of amino acids for oxidation[4] = —(15%x A)– (27% x B) | (E) Excretion of amino acids at 372mg/Kg/day[5] at the same proportions as protein composition = -($\frac{860}{2860}$ x %$AA_{Food}$ x BW x 0.372)–($\frac{2000}{2860}$ x % $AA_{body}$ x BW x 0.372) | (F) Nitrogen Balance[6] = A+B+C+D +E | (G) Excretion losses based on measured values in urine and sweat[7] = -($\frac{860}{2860}$ x %$AA_{horse\ exc}$ x BW x 0.372)–($\frac{2000}{2860}$ x % $AA_{horse\ exc}$ x BW x 0.372) | (H) Nitrogen balance[8] using measured rates of excretion F = A+B+C +D+G | (I) Nitrogen balance with increased demand for production of Hb and Myosin[9] |
|---|---|---|---|---|---|---|---|---|---|
| histidine | 21.2 | 64.8 | -59.1 | -20.7 | -5.6 | 0.6 | -3.7 | 2.5 | *-10.3* |
| serine | 44.9 | 89.1 | -93.1 | -30.8 | -8.7 | 1.4 | -13.2 | *-3.0* | *-2.8* |
| glycine | 46.7 | 211.1 | -175.7 | -64.0 | -16.8 | 1.3 | -16.2 | 1.8 | 54.5 |
| ornithine | | | | | | 0.0 | -3.4 | *-3.4* | *-3.4* |
| lysine | 34.3 | 160.2 | -132.5 | -48.4 | -12.6 | 0.9 | -8.2 | 5.4 | *-7.9* |
| threonine | 33.7 | 95.6 | -89.0 | -30.9 | -8.4 | 1.0 | -6.0 | 3.4 | 3.4 |
| valine | 47.9 | 105.5 | -106.3 | -35.7 | -10.0 | 1.5 | -6.0 | 5.5 | *-8.2* |
| leucine | 73.1 | 174.3 | -171.0 | -58.0 | -16.1 | 2.3 | -0.9 | 17.4 | 4.3 |
| isoleucine | 35.7 | 72.9 | -75.4 | -25.1 | -7.1 | 1.1 | -5.3 | 2.9 | 2.9 |
| Glutamine/ glutamic acid (Glx) | 177.8 | 233.1 | -288.8 | -89.6 | -26.7 | 5.8 | -38.3 | *-5.7* | *-19.6* |
| aspartic acid | 81.7 | 171.8 | -175.8 | -58.6 | -16.5 | 2.6 | -6.6 | 12.4 | *-1.1* |
| methionine | 13.5 | 38.3 | -35.7 | -12.4 | -3.4 | 0.4 | -1.6 | 2.2 | 2.2 |
| tyrosine | 27.8 | 57.5 | -59.2 | -19.7 | -5.6 | 0.9 | -2.2 | 4.2 | 4.2 |
| phenylalanine | 42.4 | 80.9 | -85.8 | -28.2 | -8.0 | 1.3 | -2.7 | 6.7 | *-6.1* |
| proline | 93.8 | 162.0 | -178.4 | -57.8 | -16.6 | 3.0 | -4.0 | 15.6 | 55.6 |
| alanine | 52.4 | 163.0 | -147.9 | -51.9 | -14.0 | 1.6 | -6.2 | 9.4 | 8.7 |

(*Continued*)

**Table 5.** (*Continued*)

| Amino acid (AA) calculated on the average composition in food proteins or turnover of body proteins (Table 2) | (A) Amino acids from 860g protein intake /day[1] = % $AA_{Food}$ x 860 g/day | (B) Amino acids from 2,000g endogenous protein turnover /day[2] = $\%AA_{body}$ x 2000 g/day | (C) Usage of amino acids for protein synthesis[3] = —(75%xA)— (67%xB) | (D) Usage of amino acids for oxidation[4] = —(15%x A)– (27% x B) | (E) Excretion of amino acids at 372mg/Kg/day[5] at the same proportions as protein composition = -($\frac{860}{2860}$ x %$AA_{Food}$ x BW x 0.372)–($\frac{2000}{2860}$ x % $AA_{body}$ x BW x 0.372) | (F) Nitrogen Balance[6] = A+B+C+D +E | (G) Excretion losses based on measured values in urine and sweat[7] = -($\frac{860}{2860}$ x %$AA_{horse}$ $_{exc}$ x BW x 0.372)–($\frac{2000}{2860}$ x % $AA_{horse\ exc}$ x BW x 0.372) | (H) Nitrogen balance[8] using measured rates of excretion F = A+B+C +D+G | (I) Nitrogen balance with increased demand for production of Hb and Myosin[9] |
|---|---|---|---|---|---|---|---|---|---|
| | | | | | Net balance | +25.7g | | +77.2g | |

[1]*The protein intake was set at 1.72g/Kg BW/day (860g/day for a 500Kg horse) as recommended for an animal undertaking a heavy work load [6]. The amino acid intakes (A) were extrapolated by using the average percentages of each amino acid in the feed sources (%$AA_{Food}$ ,grams of amino acids per 100 gram protein) summarised in Table 1.*

[2]*The protein turnover rate was 4 g/Kg BW/day (2,000g/day for a 500Kg horse). The amino acid availabilities from endogenous protein turnover (B) were extrapolated by using the average percentages of each amino acid in the body proteins (%$AA_{Body}$ ,grams of amino acids per 100 gram protein) summarised in Table 1.*

[3]The usage of amino acids for new protein synthesis (C) was generated by subtracting 75% of the amino acids ingested (A) and 67% of amino acids provided by protein turnover (B) [2].

[4]The usage of amino acids for oxidation (D) was generated by subtracting 15% of amino acids ingested (A) and 27% of amino acids provided by protein turnover (B) [2].

[5]The excretion losses from urine, faeces and sweat (E) were then calculated from literature values of fluid and volume losses making the assumption that the amino acids would be lost in the same ratios in which they were present in ingested protein and endogenous proteins as outlined in Table 2.

[6] The final nitrogen balance was calculated by balancing the nitrogen intake with nitrogen losses (F).

[7]*The excretion losses from urine and sweat (G) were then calculated using their measured proportions of each amino acid in sweat and urine (%$AA_{horse\ exc}$, see S1 Table).*

• The projected fluid loss by sweat for a 500Kg horse in heavy work was 5L /day [20]:

○ 46mg protein/Kg BW/day from skin and the average endogenous protein composition was used to determine losses of amino acids

○ 78mg protein /Kg BW/day sweat [9,47] and the composition of latherin was used to determine losses of amino acids in the sweat protein.

○ 5mg protein equivalents/Kg BW/day contingent of free amino acids in the fluid volume [10] and the profile of amino acids in sweat was used to determine losses of amino acids

• The urine output was calculated as 202mg protein equivalents/Kg /day and the proportions of amino acids lost in urine were determined by assessments of outputs measured in the Standardbred horses in Table 4 [9,10,44,45].

[8] The final nitrogen balance for the use of measured proportions of amino acids in sweat and urine were calculated by balancing the nitrogen intake with nitrogen losses (H).

[9] The final nitrogen balance under conditions designed to have a higher demand for the synthesis of haemoglobin and myosin (using the measured proportions of amino acids in sweat and urine) were calculated by balancing the nitrogen intake with nitrogen losses (I).

reason for this high level in horse sweat remains unclear but the rate of loss in this excretion route has an obvious negative impact on the nitrogen balance.

The negative nitrogen balance is likely to be further exacerbated for serine as it is involved in multiple metabolic pathways including synthesis of phospholipids, ceramides, sphingolipids, 5,10-methylenetetrahydrofolate (folate metabolism), cysteine and taurine[19,51]. The comprehensive rates of serine utilisation in metabolism and endogenous synthesis (for any of the amino acids) cannot be integrated into the nitrogen balance model at this stage because the information is not available. Glycine was also abundant in the sweat at 409 – 616umol/L but these levels were equivalent to or less than the corresponding levels measured in plasma [10] which suggests that the high loss of serine in sweat has a more specific role. However, the nitrogen balance of glycine is also likely to be overestimated (ie probably more negative) since, as well as comprising a third of collagen [52], it is also involved in numerous metabolic pathways including the formation of hippuric acid, purines for nucleotides and nucleic acids, haem

for haemoglobin and cytochromes, creatinine and glutathione [1]. Both of the amino acids can be synthesised by the body but rates of synthesis may not keep up with demand under certain conditions such as high intensity training and these amino acids then become conditionally essential [53–61]. Ornithine is not derived from dietary protein or endogenous protein turnover since it is not present in proteins. It is however, a vital amino acid for its role in the urea cycle for removing excess nitrogen [62–64]. It is also a precursor for arginine formation which is vital for many aspects of metabolism, including the generation of nitric oxide [65]. It must be made endogenously and is lost in both sweat and urine representing a net negative balance in this model.

The average amino acid composition in the food source for this model was estimated from six protein sources (Table 2). It was acknowledged that horses would be fed a variety of combinations of dietary protein and thus the model was run using two different combinations of rations for each protein source: legume; corn; oats; lucerne; barley; wheat. Using the sources in the rations 2:1:2:2:2:1, resulted in small changes in the average amino acid profile for this rationed mix (S2 Table). When this average profile was run in the model, minor changes in nitrogen balance were achieved with similar magnitudes of negative values observed for serine, ornithine and Glx (S3 Table). Due to the higher levels of Glx in oats and barley, which were present in higher proportions in the rationed mix, the negative nitrogen balance for Glx was reduced from -5.7g to -4.6g. A second food composition profile was tested in the rations of 1:1:1:2:2:2, which also showed slight changes in nitrogen balance but this diet regime generated a slightly more negative balance for Glx. It was concluded that the mixes of dietary protein source would have bearing on the nitrogen balance of amino acids, but Glx, serine and ornithine were most likely to be conditionally essential under intensive training conditions, primarily due their high rates of loss in excretory pathways.

It has been noted that in the racing industry, protein intake rates may actually reach levels up to 1,500g per day which are nearly double the recommendations by the NRC [6,20]. The increases are applied on the basis of matching nutritional requirements and energy for the horse under heavy or very heavy workloads [66]. On this basis, the model was run with the intake set at 3.0g/Kg BW/day to determine the impact on the nitrogen balances of the amino acids (S4 Table). The total nitrogen balance from summing the residual amino acid levels increased to 141g with serine and Glx coming into positive balance and only ornithine (-3.4g) remaining in negative balance. This higher level of protein intake has indeed brought the horse into a positive balance for all the protein amino acids. However, the question is raised that, since most of the amino acids were in positive balance, perhaps this could be more effectively achieved by targeted amino acid supplementation with a lower feed intake closer to the recommended levels of 1.72g/Kg BW/day.

The horses undergo well-defined programs to build fitness for competitive racing [67,68]. The net result can involve an increase in muscle mass and, more specifically, an enhanced capacity to deliver oxygen and remove waste $CO_2$ by augmenting red blood cell parameters. Figures showing these enhancements have been published for comparing the resting Thoroughbred with the race-fit Thoroughbred where, for example, average haemoglobin levels increased from 125g/L (resting) to 145g/L (race fit), resting haematocrit 0.40 to 0.42L/L and red blood cells 10.5 to 10.8 x$10^{12}$/L [30]. On this basis, the model was adjusted to reflect an increase in demand for haemoglobin and myosin to see whether small shifts in the rates of utilisation would have impact on the nitrogen balances of the amino acids. The minor adjustments to the average amino acid compositions of body proteins have been shown in Table 1. These new values were only used for the calculation of amino acids utilised for protein synthesis from protein turnover and ingested sources and not applied to rates of protein oxidation or rates of nitrogen excretion. The results from this scenario are shown in Table 5 in column (I). The increased rates of utilisation of the amino acids which are the major components of

haemoglobin and myosin led to substantial negative impacts on the nitrogen balances of eight amino acids, including Glx (-19.7g/day), histidine (-10.3g/day), valine (-8.2g/day), lysine (-7.9g/day), and phenylalanine (-6.1g/day).

Histidine, lysine, valine and phenylalanine are all essential amino acids that must be supplied in the diet. Histidine is required for adequate synthesis of haemoglobin and deficits can result in anaemia [69–71]. Lysine, valine and phenylalanine all have high percentage inclusions in haemoglobin, and lysine is also high in myosin. Although glutamine can be made by the body, this amino acid and lysine represent the two amino acids that have been historically used in feed supplementation programs for horses, particularly when lower quality protein sources are used for feed [7,20]. This modelling provides an explanation for why glutamine and lysine could be potentially useful as supplements for the exercising horse because they become limiting due to higher biosynthetic demands associated with exertion, generating negative nitrogen balances for these amino acids. In an earlier study, the increased levels of haemoglobin in Standardbred horses provided with a general amino acid supplementation designed to replace components lost in sweat at 30 g per day, was positively correlated ($r^2>0.98$) with the plasma levels of histidine, valine, glutamine and lysine [10].

Lawrence [66] stated that "Nutritionists and horse owners should place emphasis on meeting the amino acid requirements of the exercising horse rather than the crude protein requirements". It is proposed that the measure of specific amino acids in negative balance such as Glx and serine with additional requirements for histidine, lysine, valine and phenylalanine under conditions of fitness training to build aerobic capacity and muscle mass, identifies them as major potential limiting factors for anabolic metabolism. This would provide the framework for specific amino acid support by their supplementation to match demand during heavy and very heavy work. Even though serine, glycine, glutamine and ornithine can be synthesised endogenously, under conditions of training and high intensity performance, the body may not be able to keep up with demand [53–61,72]. These amino acids are thus proposed to represent potential key limiting factors in working horses for supporting growth and maintenance of condition due to their disproportionately high demands for use in metabolic pathways and relatively high losses via excretion pathways. If the horses were under a program of fitness development with concurrent increases in haematocrit and muscle mass, then this model would suggest that histidine, lysine, valine and phenylalanine should also be supplemented (Table 5 column (I)) with additional glutamine. The grams per day in negative balance give an indication of the daily supplementation levels to meet the requirements of the horse. Supplementation in this manner to support high intensity training and racing would enable a reduction of protein intake rates back towards the NRC recommendations.

If any amino acid is temporarily limited in supply then this could also temporarily limit protein synthesis. If free amino acids generated by exogenous turnover cannot be immediately used for protein synthesis pathways then they become subject to oxidation or conversion to fats. This could become a significant issue when additional protein is provided in the diet in an attempt to match demand of the specific amino acids, since amino acids *per se*, cannot be stored. Thus, if protein synthesis during recovery and repair from exercise was limited by the short supply of key amino acids, the rates of oxidation and/or fat conversion of the "surplus" amino acids would escalate. Once essential amino acids have been metabolised in this way they cannot be reconstructed by the horse–they must be replenished via dietary intake.

## Conclusions

It was thus concluded from the modelling exercise that horses fed at NRC recommended levels for heavy work may experience transient deficiencies at particular times in the training

program. A new supplementation strategy was proposed for the specific replenishment of these amino acids which are utilised at disproportionately faster rates during exercise. Amino acids would be best provided after exercise to replenish the potentially limiting components required to support repair and recovery. The results from the modeling provided insight as to how amino acid supplementation at quantities of 40-160mg/Kg BW/day (20-80g/day for a 500Kg horse) of the appropriate amino acids could have real benefit for maintaining a positive nitrogen balance and thus reduce demand on muscle turnover for supply. This represented 2.3–9.3% of the total protein intake for the horse which was similar to the outcome of 5 grams of amino acids per day determined for a 70Kg human, representing just 6% of the total protein intake [1]. Applied research with horses is now required to determine whether this supplementation approach can be effective in a range of breeds and competition formats. Future research should also focus on ascertaining specific amino acid requirements at various stages of training and competition to determine whether there would be long term benefits for horses such as better maintenance of muscle mass, optimising performance and reducing muscle damage.

## Supporting information

**S1 Table. The percentage relative abundances for the amino acids used in determining excretion rates/day.** The compositions for faeces and skin in sweat were taken as averages of the body proteins presented in Table 1. The urine proportions were taken from the data in Table 2. The sweat fluid abundances were derived from the data published for sweat in Standardbred horses (Dunstan et al., 2015). The sweat latherin composition was taken from the relative abundances published previously (Beeley et al., 1986). Each contribution was applied according to published rates of excretion as indicated for each component.
(PDF)

**S2 Table. The amino acid contents for each feed were adjusted to g / g food source and then multiplied based on a daily rations to derive a tailored mix.**
(PDF)

**S3 Table.** Comparisons of the excretion losses and nitrogen balances for the amino acids using (a) the average amino acids of the six food sources reported in Table 2 and (b) using the rationed mix described in Table A2.
(PDF)

**S4 Table. The adjusted nitrogen balances generated from the model under recommended feeding regime of 1.72g/Kg BW/day compared with double this rate at 3.0g/Kg BW/day for a 500Kg horse where the animal was undertaking a heavy work load and the protein turnover rate was 4 g/kg/day.**
(PDF)

**S1 Example calculation. Histidine in Table 5.**
(PDF)

## Acknowledgments

Mr Ray Harkness from Berry Park Equine (Berry Park, NSW, Australia) is thanked for his cooperation, assistance and access to horses and training facilities under his control which was vital for this project to proceed. J Franks is thanked for his role in analysing the horse urine samples for amino acid composition.

## Author Contributions

**Conceptualization:** R. Hugh Dunstan, Margaret M. Macdonald, Brittany Thorn, David Wood, Timothy K. Roberts.

**Data curation:** R. Hugh Dunstan, David Wood.

**Formal analysis:** R. Hugh Dunstan, Brittany Thorn, David Wood.

**Funding acquisition:** R. Hugh Dunstan.

**Investigation:** R. Hugh Dunstan, Margaret M. Macdonald, David Wood, Timothy K. Roberts.

**Methodology:** R. Hugh Dunstan, Margaret M. Macdonald.

**Project administration:** R. Hugh Dunstan.

**Resources:** R. Hugh Dunstan.

**Validation:** R. Hugh Dunstan, Margaret M. Macdonald, Brittany Thorn, David Wood, Timothy K. Roberts.

**Visualization:** Margaret M. Macdonald, Brittany Thorn.

**Writing – original draft:** R. Hugh Dunstan.

**Writing – review & editing:** R. Hugh Dunstan, Margaret M. Macdonald, Brittany Thorn, David Wood, Timothy K. Roberts.

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
