## [Decision Letter · Decision Letter 0]

13 Nov 2019

PONE-D-19-26715

Modelling of amino acid turnover in the horse during training and racing: A basis for developing a novel supplementation strategy

PLOS ONE

Dear Dr R. Hugh Dunstan

Thank you for submitting your manuscript to PLOS ONE. After careful consideration, we feel that it has merit but does not fully meet PLOS ONE’s publication criteria as it currently stands. Therefore, we invite you to submit a revised version of the manuscript that addresses the points raised during the review process.

We would appreciate receiving your revised manuscript by 12th December. To enhance the reproducibility of your results, we recommend that if applicable you deposit your laboratory protocols in protocols.io, where a protocol can be assigned its own identifier (DOI) such that it can be cited independently in the future. For instructions see: http://journals.plos.org/plosone/s/submission-guidelines#loc-laboratory-protocols

We look forward to receiving your revised manuscript.

Kind regards,

Tommaso Lomonaco, Ph.D

Academic Editor

PLOS ONE

Journal Requirements:

Horsepower Pty Ltd represent an Australian industry partner with an interest in developing effective supplements in the equine industry.

This company had no role in study design, data collection and analysis, decision to publish, or preparation of the manuscript.

Dr David Wood works as a veterinarian consultant to Horsepower to provide independent advice. Dr Wood assisted us in the final stages of preparation of the manuscript and provided valuable insight for checking the protein model and clinical interpretation of the data.474 The university authors have not been paid as consultants and have not received any direct grant funding from Horsepower Pty Ltd.

No - The funders had no role in study design, data collection and analysis, decision to publish, or preparation of the manuscript.

Additionally, because some of your funding information pertains to [commercial funding//patents], we ask you to provide an updated Competing Interests statement, declaring all sources of commercial funding.

In your Competing Interests statement, please confirm that your commercial funding does not alter your adherence to PLOS ONE Editorial policies and criteria by including the following statement: "This does not alter our adherence to PLOS ONE policies on sharing data and materials.” as detailed online in our guide for authors  http://journals.plos.org/plosone/s/competing-interests.  If this statement is not true and your adherence to PLOS policies on sharing data and materials is altered, please explain how.

Please include the updated Competing Interests Statement and Funding Statement in your cover letter. We will change the online submission form on your behalf.

Additional Editor Comments:

Dear Authors,

please answer all the questions arising from the reviewers comments.

Regards,

Tommaso Lomonaco

Reviewers' comments:

Reviewer's Responses to Questions

**Comments to the Author**

1. Is the manuscript technically sound, and do the data support the conclusions?

Reviewer #1: Yes

Reviewer #2: Partly

2. Has the statistical analysis been performed appropriately and rigorously? 

Reviewer #1: Yes

Reviewer #2: Yes

3. Have the authors made all data underlying the findings in their manuscript fully available?

Reviewer #1: Yes

Reviewer #2: Yes

4. Is the manuscript presented in an intelligible fashion and written in standard English?

Reviewer #1: Yes

Reviewer #2: Yes

5. Review Comments to the Author

Reviewer #1: In this work the authors' purpose was to analyse published rates of protein intake, turnover and excretion and extrapolated these to individual appraisals of amino acids in sports horses.

The work turns out to be well written, the references cited are appropriate. The division into four categories is a wright decision. The choice of two types of protein is clear and well described. The results are well described. The methods are well described.

The results are very interesting. Indeed, the results will lay important bases on understanding biochemistry of physical exercise and physiological interactions on protein turnover homeostasis. Not only that, they will also allow to better understand the type of supplement to be carried out during a training and if it is the case to carry it out.

Major revision:

1) The format result and discussion must to be divided.

Comments:

a) Could be a possibility to compare these results on possible horses supplement strategy and humans supplement?

b) Could be used The Modelling of amino acid turnover to analyse the differences into amino acid turnover in human metabolism during competition or exercise?

Reviewer #2: PLOS One Article Review

Modelling of amino acid turnover in the horse during training and racing: a basis for developing a novel supplementation strategy

Comments to Authors:

Little is known in regard to amino acid requirements in horses of all life stages; therefore, research investigating changes in amino acid turnover is an important step in determining amino acids of high priority and future dietary requirements. The objective of this study was to develop a theoretical model from previously published data to determine amino acid demand during exercise. While there are significant limitations associated with the development of this type of theoretical model, these data may still have a significant impact to the scientific community.

General Statements:

The manuscript is well-written and has the potential to positively impact the industry. While the information provided in the manuscript is valuable, the true balance of amino acids remains unclear because of their complex utilization in a multitude of metabolic and physiological pathways. I commend authors for accounting for, and acknowledging, certain variabilities and limitations; however, the data must be interpreted and utilized with caution. Ultimately, the equine industry is severely lacking the scope of research required to determine dietary requirements through a modeling study at this time. These data provide a nice mathematical indication of which amino acids should be of high priority in future amino acid research to accurately determine dietary requirements throughout various stages of life and performance intensities.

Major Concerns:

Authors do the best they can with the data available to develop a theoretical model; however, the results and conclusions must be interpreted with caution. The paper elucidates specific amino acids that should be of high priority in future animal studies thus the value to the industry; however, the industry remains off-pace in physiological research to determine dietary requirements through the use of a modelling system.

Line 421: While the model does a nice job of estimating positive and negative amino acid balance, the fed rations are variable (as stated in the paper) and not necessarily representative of all horses in heavy work. I caution authors against suggesting a specific supplementation rate due to the theoretical nature of the model and the limited biological research in amino acid nutrition, synthesis, turnover, metabolism, etc. in the horse. See general statements.

Line 440-442: I caution authors against using the verbiage “immediately after exercise”. Simply removing “immediately” from the statement would suffice.

Tables: Be consistent in formatting of tables.

Minor Concerns:

Line 148: Insert a space between 52% and of.

Line 300: Add “of” in “various types of competition”

Line 302-303: Change “initially run by assuming that…” to “initially run on the assumption that the …”

Line 312: remove “the” before urine and sweat; remove “the” before supplementary data

Line 313: Add comma after ornithine

Line 314: Add comma before respectively

Line 320: Serine is not the most abundant component of sweat, but may be the most abundant amino acid in sweat. Change “component” to “amino acid”

Line 322: Remove “the” before horse sweat fluid.

Line 323: Remove “the” before horse plasma

Line 324: Remove “the” before horse sweat

Table 5. Make sure title is above the table. It is currently split with part of the title above the table and the last line below the table.

Line 326-327: “has involvement” should be “is involved”

Line 331: “highly abundant” is redundant. Remove “highly”

Line 333: Remove “the” before plasma

Line 343: Remove “as urea”

Line 344: Arginine is involved in much more than simply NO production.

Line 376: Remove “the” before red blood cell parameters

Line 383: Remove “the” before body proteins

Line 385: Remove “the” before protein turnover

Line 392: Replace “has been shown to be” with “is”… Histidine is required for adequate synthesis…

Line 394: Add comma after haemoglobin

Line 400: Remove “supporting”

Line 420: Why have glutamine separate instead of in the list of amino acids suggested for supplementation? Instead, “this model would suggest that histidine, lysine, valine, phenylalanine, and glutamine should also be supplemented”

6. PLOS authors have the option to publish the peer review history of their article (what does this mean?). If published, this will include your full peer review and any attached files.

Reviewer #1: No

Reviewer #2: No

---

## [Author Response · Author response to Decision Letter 0]

18 Nov 2019

This has been provided as an uploaded file

---

## [Decision Letter · Decision Letter 1]

11 Dec 2019

Modelling of amino acid turnover in the horse during training and racing: A basis for developing a novel supplementation strategy

PONE-D-19-26715R1

Dear Dr. R. Hugh Dunstan,

We are pleased to inform you that your manuscript has been judged scientifically suitable for publication and will be formally accepted for publication once it complies with all outstanding technical requirements.

With kind regards,

Tommaso Lomonaco, Ph.D

Academic Editor

PLOS ONE

Additional Editor Comments (optional):

Dear Authors, the paper is ready to be published in PlosOne.

Regards,

Tommaso Lomonaco

Reviewers' comments:

Reviewer's Responses to Questions

**Comments to the Author**

1. If the authors have adequately addressed your comments raised in a previous round of review and you feel that this manuscript is now acceptable for publication, you may indicate that here to bypass the “Comments to the Author” section, enter your conflict of interest statement in the “Confidential to Editor” section, and submit your "Accept" recommendation.

Reviewer #2: All comments have been addressed

2. Is the manuscript technically sound, and do the data support the conclusions?

Reviewer #2: Yes

3. Has the statistical analysis been performed appropriately and rigorously? 

Reviewer #2: Yes

4. Have the authors made all data underlying the findings in their manuscript fully available?

Reviewer #2: Yes

5. Is the manuscript presented in an intelligible fashion and written in standard English?

Reviewer #2: Yes

6. Review Comments to the Author

Reviewer #2: (No Response)

7. PLOS authors have the option to publish the peer review history of their article (what does this mean?). If published, this will include your full peer review and any attached files.

Reviewer #2: No

---

## [Editor Report · Acceptance letter]

13 Dec 2019

PONE-D-19-26715R1 

Modelling of amino acid turnover in the horse during training and racing: A basis for developing a novel supplementation strategy 

Dear Dr. Dunstan:

I am pleased to inform you that your manuscript has been deemed suitable for publication in PLOS ONE. Congratulations! Your manuscript is now with our production department. 

With kind regards,

on behalf of

Dr. Tommaso Lomonaco 

Academic Editor

PLOS ONE